# Characterization of a Thermostable and Surfactant-Tolerant Chondroitinase B from a Marine Bacterium *Microbulbifer* sp. ALW1

**DOI:** 10.3390/ijms23095008

**Published:** 2022-04-30

**Authors:** Mingjing Mou, Qingsong Hu, Hebin Li, Liufei Long, Zhipeng Li, Xiping Du, Zedong Jiang, Hui Ni, Yanbing Zhu

**Affiliations:** 1College of Ocean Food and Biological Engineering, Jimei University, Xiamen 361021, China; M2427634823@163.com (M.M.); hqs3073186436@163.com (Q.H.); 201911832018@jmu.edu.cn (L.L.); lzp2019@jmu.edu.cn (Z.L.); xipingdu@jmu.edu.cn (X.D.); zdjiang@jmu.edu.cn (Z.J.); nihui@jmu.edu.cn (H.N.); 2Department of Pharmacy, Xiamen Medical College, Xiamen 361023, China; lhb@xmmc.edu.cn

**Keywords:** chondroitinase B, *Microbulbifer* sp., thermostable, surfactant-tolerant

## Abstract

Chondroitinase plays an important role in structural and functional studies of chondroitin sulfate (CS). In this study, a new member of chondroitinase B of PL6 family, namely ChSase B6, was cloned from marine bacterium *Microbulbifer* sp. ALW1 and subjected to enzymatic and structural characterization. The recombinant ChSase B6 showed optimum activity at 40 °C and pH 8.0, with enzyme kinetic parameters of *K*_m_ and *V*_max_ against chondroitin sulfate B (CSB) to be 7.85 µg/mL and 1.21 U/mg, respectively. ChSase B6 demonstrated thermostability under 60 °C for 2 h with about 50% residual activity and good pH stability under 4.0–10.0 for 1 h with above 60% residual activity. In addition, ChSase B6 displayed excellent stability against the surfactants including Tween-20, Tween-80, Trion X-100, and CTAB. The degradation products of ChSase B6-treated CSB exhibited improved antioxidant ability as a hydroxyl radical scavenger. Structural analysis and site-directed mutagenesis suggested that the conserved residues Lys248 and Arg269 were important for the activity of ChSase B6. Characterization, structure, and molecular dynamics simulation of ChSase B6 provided a guide for further tailoring for its industrial application for chondroitin sulfate bioresource development.

## 1. Introduction

Glycosaminoglycans (GAGs) are negatively-charged linear heteropolysaccharides that are ubiquitously present on cell surfaces and in extracellular matrices of animals [1]. GAGs are structurally diverse, including classes of chondroitin sulfate/dermatan sulfate (CS/DS), hyaluronic acid (HA), heparin/heparan sulfate (Hep/HS), and keratan sulfate (KS) [2]. CS is a polymer of the basic repeating disaccharide unit composed of β-1⟶3-linked glucuronic acid (GlcA) and N-acetylgalactosamine (GalNAc) [3]. The sulfation modification at various positions leads to the heterogeneity of CSs and produces some major forms of CS including CS-O, A, B, C, D, E, F, and K [4]. The bioavailability of CSs has been questioned due to their high molecular weight, while their degradation products CS oligosaccharides have been proved to possess higher bioactivity and bioavailability [5,6,7]. The preparation of low-molecular-weight CS can be achieved through chemical [8], physical [9], and enzymatic methods [10,11]. The enzymatic approach is normally more favored than the chemical or physical approaches as the enzymatic digestion is more under control to produce desired oligos in an environmentally friendly manner [10,12].

Chondroitinases (ChSases) are the enzymes that digest the CS chain to generate different repeats of disaccharides and can be categorized into hydrolase and lyase in light of their enzymatic mechanism. In contrast to animals that utilize hydrolase to break down CS polysaccharide, microorganisms utilize lyase via the so-called β-elimination mechanism to cleave the CS chain [13,14,15]. Based on their substrate specificity, ChSases can be divided into different types including ChSase ABC, ChSase AC, ChSase B, and ChSase C. In addition to their application for preparing biologically active CS oligosaccharides [16], ChSases can also serve as a useful tool for dissection of the structure and function of CS [13,16,17,18,19] and for analysis and treatment of some diseases [17,20,21].

A marine bacterium *Microbulbifer* sp. ALW1 was isolated and characterized as a potent strain capable of degrading brown algae cell wall [22]. Through functional genomic analysis of strain ALW1, a putative gene ChSase B6 encoding a ChSase B was identified, and the enzymatic properties and structure of ChSase B6 were characterized. This study would provide a guide for developing this CS-cleaving enzyme for CS-related research and application.

## 2. Results and Discussion

### 2.1. Sequence Analysis of Chondroitinase

The gene encoding ChSase B6 contained an open reading frame of 2277 bp for a protein of 758 amino acid residues, with theoretical molecular mass and pI values 82.5 kDa and 6.40, respectively. Protein domain structure showed an N-terminal domain (NTD, Val^36^-Gly^404^) and a C-terminal domain (CTD, Glu^479^-Asn^589^) in ChSase B6 and signified a member of the polysaccharide lyase 6 (PL6) family. ChSase B6 shared the highest sequence identity of 87.28% to WP_197023865.1, a putative alginate lyase from *Microbulbifer* sp. HZ11. Phylogenetic analysis of ChSase B6 supported that the enzyme belonged to PL6 family as it formed a distinct group with the PL6 family members (Figure 1). The sequence similarity between ChSase B6 and the characterized chondroitinase from *Pedobacter heparinus* of PL6 family was 25.35%.

### 2.2. Chondroitinase Expression and Substrate Specificity

To obtain ChSase B6 protein for enzyme activity assay, it was heterologously expressed in *E. coli*, and the purified protein was examined with SDS-PAGE. A recombinant protein with 86.2 kDa was detected after induction with IPTG (Figure 2A). High-resolution tandem mass spectrometry (MS/MS) has been widely used to identify peptides, and proteins identified by MS/MS can be reconstructed from the peptide data [23]. In this study, the protein band recovered from the gel was tested by tandem mass spectrometry, and 40 peptide sequences identified covering 35% of the full length of ChSase B6 further confirmed the specificity of protein expression and purification (Table 1).

The activity profile of ChSase over different substrates indicated that it exhibited activity against CSB and HA, while it had almost negligible activity against CSA and CSC (Figure 2B). The activity of ChSase B6 against HA was comparatively lower than that against CSB (32.8% relative activity), indicating that ChSase B6 had preference for CSB. Similar results of substrate specificity were observed in chondroitinases B from *Flavobacterium heparinum* [24].

### 2.3. Enzymatic Properties of ChSase B6

The plotting of enzyme activity over the incubation temperature demonstrated that ChSase B6 displayed its optimal activity at 40 °C (Figure 3A), which was higher than the other chondroitinases listed in Table 2. In addition, ChSase B6 could function effectively over the temperatures ranging from 4 to 50 °C, exhibiting over 60% of the maximum activity (Figure 3A). When the temperature reached 60 °C, the enzyme had 52.3% of its maximum activity (Figure 3A). After prolonged incubation at 30 °C, ChSase B6 gradually decreased in residual activity, dropping from 80.4% at 2 h to 59.1% at 10 h (Figure 3B). When the incubation temperature was elevated to 40 °C, ChSase B6 retained 68.2% and 34.0% residual activities at 2 h and 10 h, respectively (Figure 3B). When the temperature was raised to 50 °C and 60 °C, ChSase B6 activity was almost abolished after incubation for 10 h, while retaining 61.0% and 49.7% of residual activities at 2 h, respectively (Figure 3B). ChSase B6 exhibited a superior thermostability compared to marine *Vibrio* sp. QY108 derived Chondroitinase ABC (Table 2).

Enzyme activity assay for ChSase B6 under different pH conditions showed that ChSase B6 had optimum activity at pH 8.0 (Figure 3C), the same as the chondroitinases from *Pedobacter heparinus* and *Flavobacterium heparinum* (Table 2). It displayed over 55% of its maximal activity under suboptimal pH at 7.0–10.0 (Figure 3C). The results of pH stability assay indicated that ChSase B6 could tolerate the pH ranging from 4.0 to 10.0, with the residual activity above 60% after exposure (Figure 3D). When the treatment pH increased to 11.0, the enzyme activity diminished abruptly, with 27.8% residual activity (Figure 3D). These results suggested that ChSase B6 could be applied in a wide range of pH conditions. Under the optimum temperature and pH conditions, the kinetic parameters *K*_m_ and *V*_max_ of ChSase B6 against CSB substrate were determined to be 7.85 µg/mL and 1.21 U/mg, respectively.

### 2.4. Effects of Chemicals on ChSase B6 Stability

Metal ions might be actively involved in regulating enzyme catalytic action or coordinating enzyme conformational shape during the interaction with substrate, thus affecting enzyme activity [29]. At the tested concentrations, Na^+^ and K^+^ did not show significant impact on ChSase B6 activity. Other tested metal ions, including Ag^+^, Ba^2+^, Ni^2+^, Mn^2+^, Hg^2+^, Fe^2+^, Cu^2+^, Ca^2+^, Mg^2+^, Zn^2+^, and Fe^3+^, substantially inhibited the enzymatic activity, with the most pronounced inhibition from Cu^2+^ and Fe^3+^ (Figure 4A).

Upon exposure to other chemical additives examined, the activity of ChSase B6 was slightly inhibited by the chelating agent ethylenediaminetetraacetic acid (EDTA) at 10 mM, while stimulated by the reducing agents dithiothreitol (DTT) and β-mercaptoethanol (β-ME) especially at 10 mM (Figure 4B). The stimulatory effect of DDT and β-ME indicated that the maintenance of the reducing status of the cysteine residues in ChSase B6 was beneficial to its enzymatic activity. The nonionic surfactant Trion X-100 displayed marginal influence on the enzyme activity at 1%, with a 21.5% increase in enzyme activity at the concentration of 0.1%. Other nonionic surfactants, including Tween-20 and Tween-80, enhanced ChSase B6 activity at the concentrations of 0.1% and 1%. The cationic surfactant cetyltrimethylammonium bromide (CTAB) increased ChSase B6 activity by 25.8% at the concentration of 1%, while the anionic surfactant sodium dodecyl sulfate (SDS) inhibited the enzyme activity by 87.6% at the same concentration. It is reported that the nonionic surfactants can be considered as more benign to an enzyme than the ionic ones as a general rule, and this difference can be attributed to the difference in binding mode [30]. The high stability of ChSase B6 to some surfactants would benefit its industrial utilization. In addition, the denaturants urea and guanidine hydrochloride (GdnHCl) showed significant and moderate negative effects on the enzyme activity, with 91.1% and 54.5% inhibitions at 1 M, respectively (Figure 4B).

### 2.5. Hydroxyl Radical Scavenging Activity of the Enzymatic Products

The degradation products of CSB digested by ChSase B6 was analyzed by mass spectrometry. In the positive ion mode, the mass-to-nucleus ratio of the product at 527.4 *m/z* [M+Na+H]^+^ corresponds to disaccharide, where M is the molecular weight of the neutral disaccharide (2-acetamido-2-deoxy-3-*O*-(β-D-gluco-4-enepyranosyluronic acid)-4,6-di-*O*-sulfo-D-galactose) completely saturated with sodium ions [31]. MS results showed that ChSase B6-mediated degradation of CSB produced disaccharide (Figure 5A). The antioxidant activity of the degradation products of CSB by ChSase B6 was evaluated by hydroxyl radical scavenging assay. Hydroxyl radicals are the most active oxygen free radicals and can induce nearby oxidized molecules to suffer serious oxidative damage [32]. The hydroxyl radical scavenging activity of the enzymatic products was markedly improved with the increase in its concentration compared to the undigested CSB (Figure 5B). These results suggested that the digestion by ChSase B6 could significantly boost the antioxidant activity of CSB polymer.

### 2.6. CD Spectroscopy of ChSase B6

The secondary structure of ChSase B6 was estimated by CD analysis. The amounts of helix, antiparallel, parallel, β-turn and random coil were about 10.6%, 38.1%, 4.5%, 15.8%, and 31.3%, respectively. The results indicated the antiparallel and random coil structures were predominant in ChSase B6.

### 2.7. Structure Modeling

The three-dimensional structure of ChSase B6 was modeled with an alginate lyase AlyGC from marine bacterium *Glaciecola chathamensis* S18K6^T^ (PDB entry 5GKD) as a template. The model covered a nearly full-length sequence of ChSase B6 (amino acid residues 30–754), sharing 58.48% of sequence identity. The overall structure of ChSase B6 was predicted to fold into a “tower-like” structure, and the two domains of NTD and CTD adopted the β-helix fold (Figure 6A). ChSase B6 exhibited similar topology with Chondroitinase B (ChonB) from *Flavobacterium heparinum* (PDB number: 1DBO). It is worth mentioning that ChSase B6 differed from ChonB with a tandem β-helix fold instead of a single right-handed parallel β-helix, and it was absent an α-helix in the catalytic cleft (Figure 6B). It has been reported that the α-helix in ChonB is important for its interaction with substrate and catalytic function [26,33].

The structural superposition of ChSase B6 and ChonB showed that most of residues superposed well with the active sites of ChonB with C_α_ root mean square deviation (RMSD) of 1.321 Å, except for Arg182, Val242, Phe314, Arg331, and Tyr332 in ChSase B6 (Figure 6C). It was speculated that the active sites of ChSase B6 were composed of Arg182, Asn209, Glu241, Val242, Glu243, Lys248, Arg269, His270, Phe314, Arg331, and Tyr332 as their counterparts in 1DBO [26]. Among them, Lys248 and Arg269 of ChSase B6 was predicted to function as the Brønsted base and Brønsted acid in the catalysis reaction, respectively. Substitutions of Lys248 and Arg269 by alanine resulted in relative activities of 51.8% and 48.7%, respectively, suggesting that the two conserved residues were important for the enzymatic activity of Chase B6. In addition, the catalytic cleft of ChonB (Figure 6E) is more L-like compared with that of ChSase B6 (Figure 6D) in this study.

### 2.8. Thermal Adaptation Analysis by MD Simulation

MD simulation can provide information on the global structure changes of the protein when exposed to different temperatures, reflecting the thermal motion of the protein structure. RMSD can monitor the differences between the back-bones of a protein from its initial structural conformation to its final position [34]. Using the initial structure of ChSase B6 as a reference, the RMSD values in 20 ns MD simulations were calculated (Figure 7A). The results showed that ChSase B6 achieved stable states with average RMSDs of 0.28 Å, 0.27 Å, and 0.33 Å at temperatures of 303 K (30 °C), 313 K (40 °C), and 323 K (50 °C), respectively, suggesting that higher temperature caused an increase in the overall enzyme structural fluctuations. RMSF values of MD simulations showed that different protein domains of ChSase B6 responded differently to temperature changes (Figure 7B). Taking RMSF values of the enzyme at 303 K as a reference, the regions of increased protein flexibility at 323 K were presented (Figure 7B,C). Notably, residues in the loop region (324–333) near the enzymatic activity pocket and the junction (402–504) between NTD and CTD of the enzyme showed higher flexibility with increasing temperature. The structural stabilization of proteins generally relies on rigidifying features such as salt bridges, disulfide bonds, and hydrophobic interactions [35]. The van der Waals bonds were detected between the residues of S222 and Y332 and the catalytic residues K248 and R269, respectively (Figure 7C), and the increased flexibility of residues S222 and Y332 was bound to affect the thermostability and catalytic activity of ChSase B6. Rigidifying flexible sites has been proven to be an effective approach to improving the thermostability of chondroitinase [36]. MD simulation results in this study could provide useful information to improve the stability and enzymatic activity of ChSase B6.

## 3. Materials and Methods

### 3.1. Sequence and Structure Analysis

The nucleotide sequence of ChSase B6 (GenBank accession number OK626321) was isolated from *Microbulbifer* sp. ALW1 genome (GenBank accession number CP047569) after genomic sequence analysis. Homolog and conserved domain (CD) searches were performed on NCBI platform using BLAST and CD-search tools, respectively. Signal peptide was predicted using SignalP 5.0. A phylogenetic tree was constructed by neighbor-joining (NJ) method using MEGA 7.0 software. The protein structure was computed with the SWISS-MODEL, and was visualized and analyzed by the PyMOL Molecular Graphics System (DeLano Scientific LLC, San Carlos, CA, USA). The online tool RING-2.0 [37] (http://protein.bio.unipd.it/ring; accessed on 21 October 2021) was used to analyze the interactions around catalytic residues in this work. MD simulation (20 ns) was carried out using GROMACS 5.1.4 (http://www.gromacs.org/; accessed on 1 February 2022) in conjunction with GROMOS96 54a7 force field [38] accordingly to the method described by Li et al. [39]. The root means square deviation (RMSD) and the root means square fluctuation (RMSF) were calculated using GROMACS rms and rmsf tools. Postprocessing were performed using standard GROMACS tools.

### 3.2. Recombinant Construction and Protein Expression

The gene encoding ChSase B6 excluding signal peptide was cloned into pET-28a vector following conventional cloning strategy. The recombinant plasmid pET-28a-*ChSase* was introduced into *E. coli* BL21 (DE3) for ChSase B6 expression. Protein expression was performed with reference to the procedures described [40]. The induced cells were harvested by centrifugation at 7000× *g* for 10 min. Under the native conditions, Ni sepharose 6 Fast Flow (GE Healthcare Life Sciences) affinity chromatography was conducted to purify the His-tagged protein. The cell pellet was resuspended in 12 mL of binding buffer (50 mM NaH_2_PO_4_, 300 mM NaCl, 20 mM imidazole, pH 8.0). After the cells were lysed by sonication, the lysate was centrifugated at 12,000× *g* for 20 min at 4 °C. The resins were added to the harvested cleared lysate. After reaction at 4 °C for 25 min with shaking, the lysate–resins mixture was loaded in a column. The resins were washed with wash buffer (50 mM NaH_2_PO_4_, 300 mM NaCl, 40 mM imidazole, pH 8.0). The binding protein was eluted with elution buffer (50 mM NaH_2_PO_4_, 300 mM NaCl, 250 mM imidazole, pH 8.0), and the eluate was dialyzed against 50 mM Tris-HCl buffer (pH 8.0). The purified protein was resolved on SDS-PAGE to determine its molecular weight and homogeneity and was quantified with bicinchoninic acid (BCA) protein assay kit (Pierce, WA, USA). Protein identity was analyzed by nano-electrospray ionization mass spectrometry/mass spectrometry (Nano-ESI MS/MS) (Waters, Milford, MA, USA) after cleanup from acrylamide gel. Protein identification was performed with Protein Pilot™ v4.5 software (AB SCIEX, Framingham, MA, USA) against the Malus×domestica database (http://www.rosaceae.org; accessed on 3 June 2021) using the Paragon Algorithm and searching against the genome database of *Microbulbifer* sp. ALW1 [41].

### 3.3. Circular Dichroism (CD) Spectroscopy

The purified protein was dialyzed against deionized water, and analyzed using a circular dichroism (CD) spectrometer (Applied Photophysics Ltd., Leatherhead, UK) at room temperature in the wavelength range of 185–260 nm. The operating parameters of scan rate, interval, and bandwidth were 100 nm/min, 0.25 s, and 1.0 nm, respectively.

### 3.4. Enzyme Activity Assay

The assay of enzyme activity was performed at 40 °C by incubating 15 µL of the purified recombinant enzyme (0.6 mg/mL) with 680 µL of 2 mg/mL chondroitin sulfate B (Soiarbio, China) substrate solution (in 50 mM sodium phosphate buffer, pH 8.0) for 20 min. After heat inactivation, the reaction was allowed to cool to room temperature. The UV absorption at 232 nm was measured using a spectrophotometer (Shanghai Metash Instruments Co., Ltd., Shanghai, China). One unit of the enzyme activity was defined as the amount of enzyme that produced 1 µmol of unsaturated carbon bonds per min at the above conditions.

### 3.5. Substrate Specificity

The enzyme activity against different substrates including CSA, CSB, CSC, and HA (Soiarbio, Beijing, China) was examined to determine the substrate specificity. The enzyme activities over these substrates were determined as mentioned above.

### 3.6. Enzyme Property Assay

The impact of temperature on enzyme activity was determined by measuring the enzyme activity at different temperatures (4–60 °C). The impact of temperature on enzyme stability was determined by measuring the residual activity after exposure to different temperatures for 2–10 h. The enzyme activity without heat treatment was defined as 100%. The effect of pH on enzyme activity was monitored by scoring the enzyme activity under different pH conditions (4.0–11.0). The effect of pH on enzyme stability was monitored by scoring the residual activity after incubation the enzyme under varying pH at 37 °C for 1 h. The enzyme activity without pH treatment was defined as 100%. Enzyme parameters of *V*_max_ and *K*_m_ were calculated by plotting the enzyme activity over 0.01–0.5 mg/mL CSB.

### 3.7. Effects of Chemicals on Enzyme Stability

The influence of different metal ions and other chemicals (including the chelating and reducing agents, surfactants, and denaturants) on enzyme stability was evaluated by assaying the residual activity after exposure the enzyme to the additives at specified concentrations at 37 °C for 1 h. The enzyme activity for the treatment was normalized to the control without any addition (100%).

### 3.8. Preparation and Identification of Enzymatic Products

The degradation products of CSB by ChSase B6 were prepared by initial digestion of 100 mL of 5 mg/mL CSB with 100 U of ChSase B6 at 40 °C for 3 h followed by another 3 h of incubation after supplemented with additional 30 U of enzyme. The digestion progress, monitored at every 30 min, indicated that the product amounts reached the steady state. After heat inactivation and centrifugation, the product supernatant was harvested and applied on a Millipore centrifugal filter 3K device (Millipore, Burlington, MA, USA). The filtrate was lyophilized, and then analyzed by TSQ QUANTUM ACCESS MAX liquid chromatography-mass spectrometer (Thermo, Waltham, MA, USA).

### 3.9. Hydroxyl Radical Scavenging Assay

The lyophilized CSB enzymatic products were reconstituted with deionized water to generate a concentration gradient. Hydroxyl radical scavenging activity of the samples with different concentrations was examined using the method described before [22]. In brief, the reaction solution was composed of 0.2 mL of the sample solution, 0.1 mL of 9 mM salicylic acid (dissolved in ethanol), 0.1 mL of 8.8 mM H_2_O_2_, 0.1 mL of 9 mM FeSO_4_, and 0.6 mL H_2_O. After incubation at 37 °C for 10 min, the absorbance at 510 nm was determined. The hydroxyl radical scavenging effect of the sample was calculated as follows:$$\mathrm{Scavenging}\;\mathrm{effect}\;(\%)=(1-\frac{A_{1}}{A_{0}})\;\times \;100$$
where *A*_0_ is the absorbance of the control without sample and *A*_1_ is the absorbance of the sample.

### 3.10. Site-Directed Mutagenesis

Site-directed mutagenesis of ChSase B6 at Lys248 and Arg269 residues was performed using a commercial kit (TaKaRa, Kusatsu, Japan) following the mutagenesis protocol. The activity of the mutants relative to the wide type enzyme (WT) was calculated.

## 4. Conclusions

In this study, ChSase B6, a chondroitinase B of PL6 family, was identified from the marine bacterium *Microbulbifer* sp. ALW1 and characterized for its enzymatic parameters and structural modeling. ChSase B6 could effectively digest CSB substrate into disaccharides with optimum activity at 40 °C and pH 8.0. ChSase B6 demonstrated good thermostability and pH stability. ChSase B6 exhibited excellent stability against the tested non-ionic surfactants and the cationic surfactant CTAB. The fragmentation of CSB by ChSase B6 offered improved antioxidant ability as a hydroxyl radical scavenger. The modeled structure and MD simulation of ChSase B6 provided an insight into the molecular basis of enzyme activity and thermostability, serving as a guide for protein engineering. The activity and stability of ChSase B6 made it a competitive candidate for further fundamental research, as well as medical, industrial, and biotechnological applications.

## Figures and Tables

**Figure 1 ijms-23-05008-f001:**
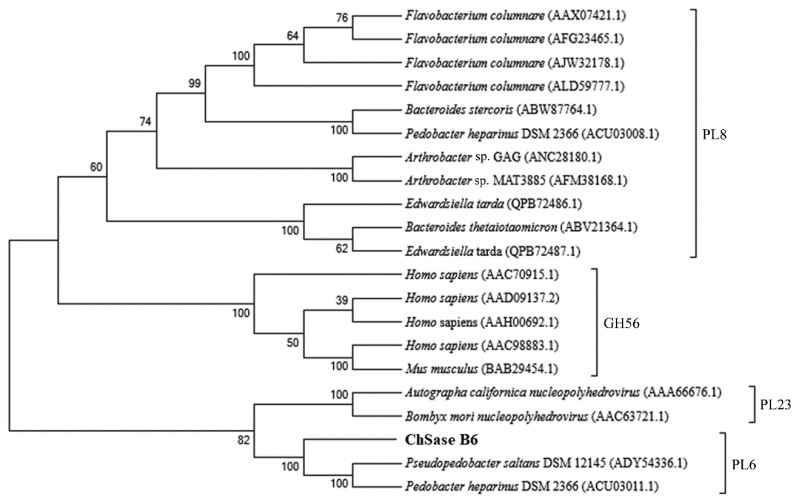
Phylogenetic analysis of ChSase B6. The phylogenetic tree was constructed with MEGA 7.0 using the neighbor-joining method.

**Figure 2 ijms-23-05008-f002:**
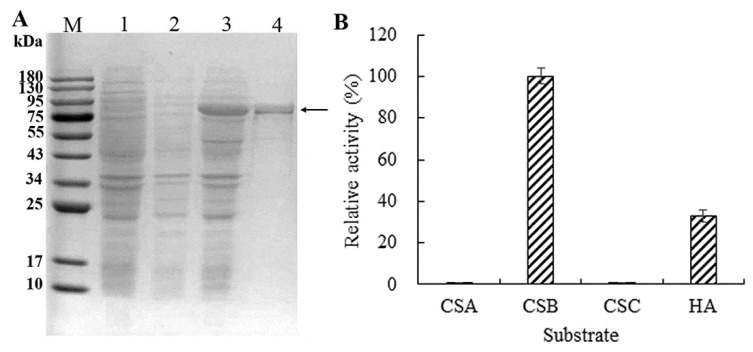
Expression and substrate specificity of recombinant ChSase B6. (**A**) Expression of the recombinant ChSase B6 in *E. coli.* Lane M: protein molecular weight marker; Lane 1: induced *E. coli* containing pET-28a(+); Lane 2: non-induced *E. col**i* containing pET-28a-*ChSase*; Lane 3: induced *E. col**i* containing pET-28a-*ChSase*; Lane 4: purified recombinant ChSase B6 protein; (**B**) Substrate specificity of recombinant ChSase B6. The substrates tested include chondroitin sulfate A (CSA), chondroitin sulfate B (CSB), chondroitin sulfate C (CSC), and hyaluronic acid (HA).

**Figure 3 ijms-23-05008-f003:**
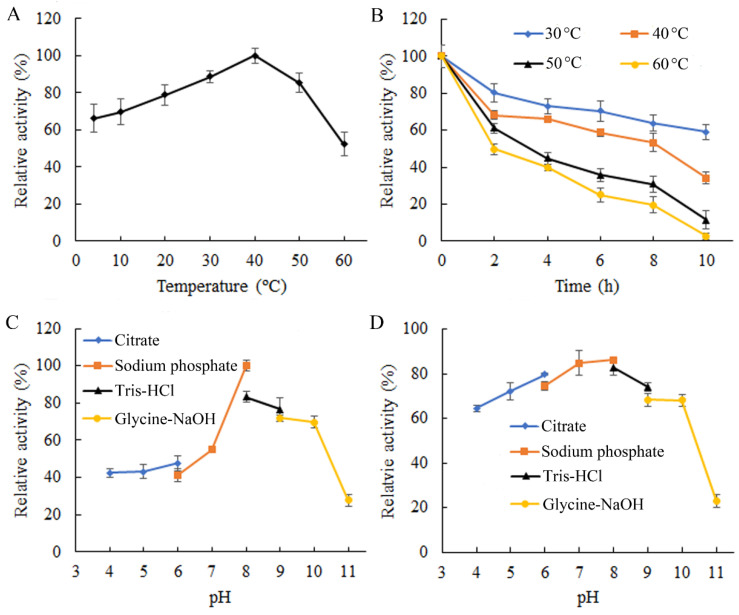
Enzymatic properties of ChSase B6. (**A**) Enzymatic activity of ChSase B6 at different temperatures. Activity at 40 °C was taken as 100%; (**B**) thermostability of ChSase B6. The enzyme activity without heat treatment was taken as 100%; (**C**) enzymatic activity of ChSase B6 under different pH conditions. Activity at pH 8.0 was taken as 100%; (**D**) pH stability of ChSase B6. The enzyme activity without pH treatment was taken as 100%. Each value represents the mean of three replicates ±SD.

**Figure 4 ijms-23-05008-f004:**
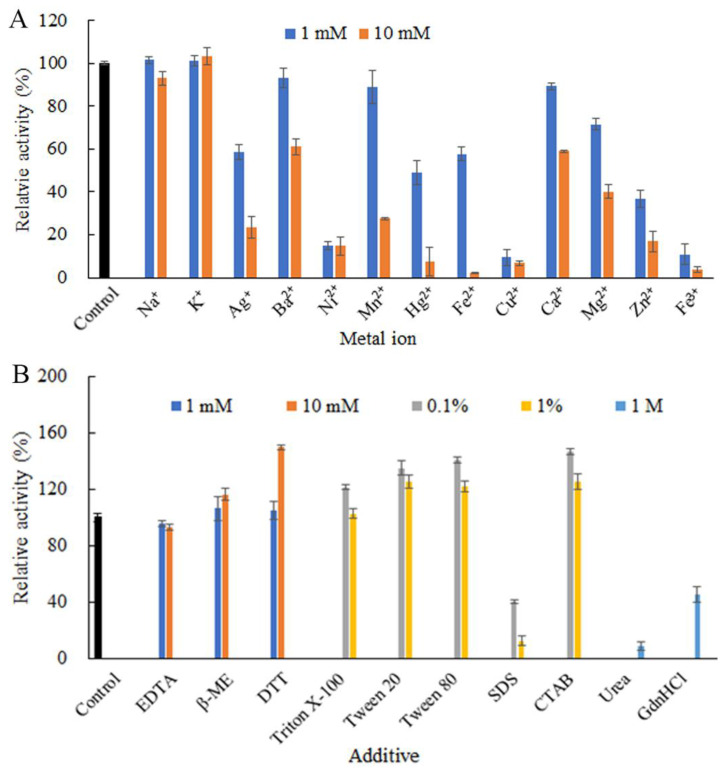
Effects of chemical additives on the stability of ChSase B6. (**A**) Effects of metal ions on the recombinant ChSase B6 stability; (**B**) effects of chelating agents, reducing reagents, surfactants, and denaturants on ChSase B6 stability. Values are mean ± SD from three independent experiments.

**Figure 5 ijms-23-05008-f005:**
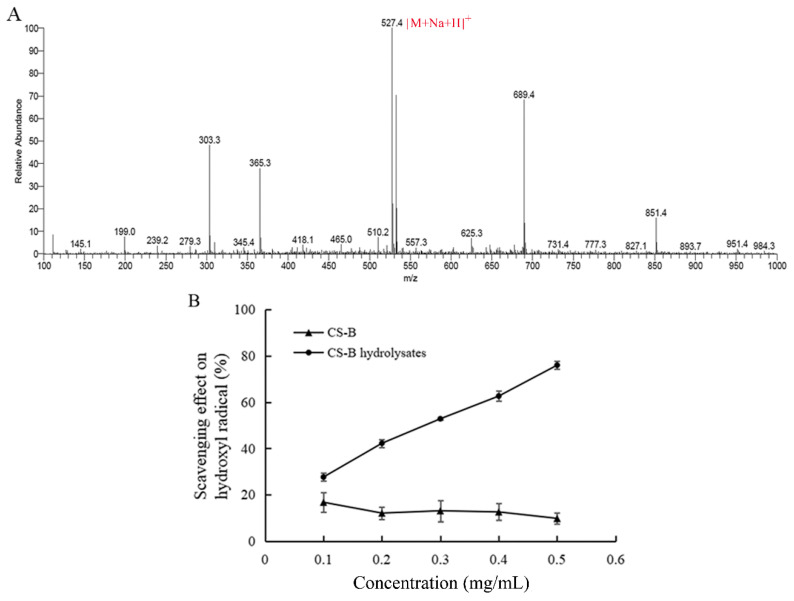
Identification and antioxidant activity of the enzymatic products. (**A**) MS analysis of CSB degradation products by ChSase B6; (**B**) hydroxyl radical scavenging activity of CSB degradation products by ChSase B6. Values are presented as mean ± SD from three independent experiments.

**Figure 6 ijms-23-05008-f006:**
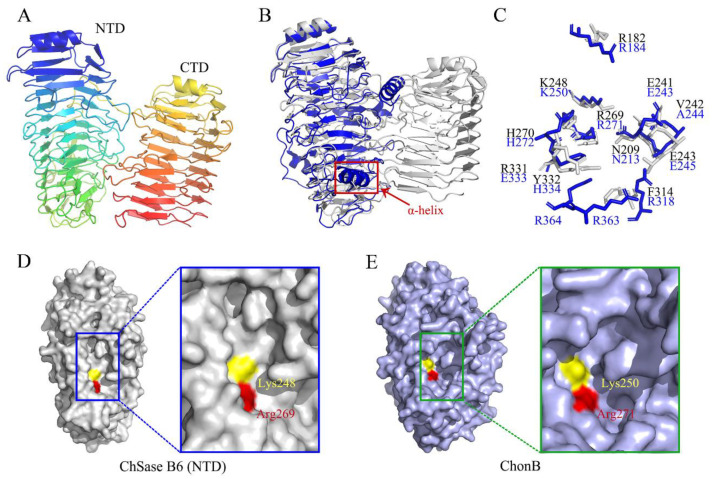
Structural comparison of ChSase B6 and ChonB. (**A**) Overall structural of ChSase B6. (**B**) Structural alignment of ChSase B6 and ChonB. ChSase B6 is colored in gray. ChonB is shown in blue schematic view and the α-helix is circled. (**C**) Partial superposition of ChSase B6 and ChonB. Residues of ChSase B6 and ChonB are shown in gray and blue, respectively. (**D**) Surface views of the NTD of ChSase B6. (**E**) Surface views of ChonB. The catalytic clefts are circled. Brønsted bases and acids are colored in yellow and red, respectively.

**Figure 7 ijms-23-05008-f007:**
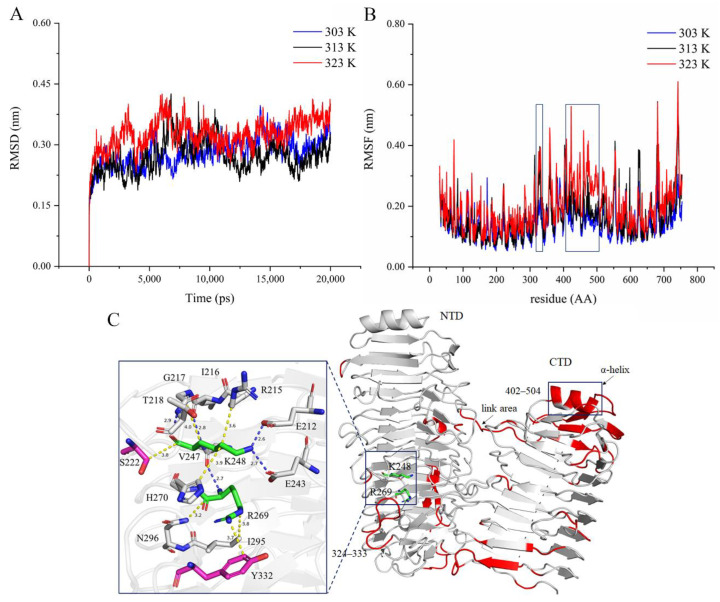
ChSase B6 at different simulation temperatures. (**A**) RMSD of ChSase B6 at 303 K (30 °C), 313 K (40 °C), and 323 K (50 °C); (**B**) RMSF of ChSase B6 at 303 K (30 °C), 313 K (40 °C), and 323 K (50 °C). (**C**) The enzyme flexibility changes from the simulation temperature 303K to 323K. Red is the area of increased flexibility. In the residue interaction diagram, the bonds in yellow and blue represent van der Waals forces and hydrogen bonds, respectively.

**Table 1 ijms-23-05008-t001:** Tryptic peptide sequencing of fractionated ChSase B6 by Nano-ESI-MS/MS.

Sequence	Length (aa)	Position (aa)	Score	Mass (Da)	Fraction Average	Retention Time (Min)
KLEAGDSVR	9	48–56	99	487.7671	2	6.95
LANGVWK	7	57–63	96.27	394.7161	2	17.56667
DFEILFTGNGEK	12	64–75	99	685.8295	2	5.36 × 10^0.01^
VILSGQSNLR	10	89–98	98.61	544.3088	2	19.11667
LAGDHLVVSGLVFK	14	99–112	99	485.617	3	44.71667
DHLANHSRVTEVVIEDFSKPER	22	129–150	32.56	868.4452	3	67.6
MESDYWVGIY	10	151–160	99	631.7729	2	61.46667
FDHSHLAGK	9	166–174	99	506.2507	2	7.133333
GVTMAVR	7	178–184	99	375.2045	2	7.016667
LDSEQSQENR	10	184–193	99	603.2728	2	5.333333
PVLGSNGGETLR	12	206–217	99	600.814	2	19.96667
IGTSHYSLTDSMTLVENNFFDR	22	218–239	99	855.0704	3	50.56667
NNTFYESR	8	258–265	99	516.2231	2	16.15
IFLGNGVDHTGGIR	14	282–295	99	728.8798	2	28.83333
VINADQVIR	9	296–304	99	535.801	2	40.43333
NNYLEGLTGYR	11	305–315	61.11	572.2689	2	51.91667
FGSGF	5	316–320	25.78	514.2279	1	24.9
TVMNGVPNSPINR	13	321–333	99	700.352	2	22.88333
YHQVVNAK	8	334–341	99	479.7429	2	5.85
IEHNSFINVEHIYL	14	342–355	79.68	864.4382	2	49.95
DSTISDNIFYTNNGK	15	369–383	99	845.3834	2	38.58333
SPFSVFDDISGITF	14	384–397	98.53	766.3783	2	70.1
AKNGLLYPIDSAAASK	16	424–439	99	810.4335	2	29.9
DFEILFTGNGEK	12	449–460	99	685.8273	2	53.75
IHGGINK	7	450–456	18.2	369.7809	2	12.733
EMTGVSWYPK	10	463–472	99	599.2839	2	33.36666
SEPVTPFDSGK	11	473–483	99	582.2825	2	18.85
AIAGAADGDTLMLENGTYNARK	22	497–518	24.37	708.9996	3	37.35
TAVWGMLK	8	570–577	95.37	777.3964	1	60.08333
FSMSNSR	7	581–587	97.18	415.1802	2	11.58333
VENLDINHSYHFFDSGHR	18	588–605	99	730.0019	3	32.3
ISLSNNL	7	612–618	98.04	380.7134	2	32.15
EVVIEDFSKPER	12	619–630	99	724.3733	2	32.1
NITGDLLK	8	632–639	99	437.2512	2	26.7
EQDDLGIYNAE	11	643–653	35.16	633.7799	2	32.41667
TFDQVQGALVDLYR	14	661–674	99	812.9194	2	54.75
GGTDESTFGPHIDFSK	16	675–690	99	565.5933	3	32.38334
NKAQALIK	8	702–709	25.37	414.7721	2	17.31667
LHGTQVADIQDNR	13	711–723	99	489.5826	3	10.81667
APLPEVTELVAEGPHTAVLANNEVTN	26	750–775	99	895.7964	3	5.87 × 10^0.01^

**Table 2 ijms-23-05008-t002:** The enzymatic properties of some chondroitinases of bacterial origin.

Source	Substrate Specificity	Optimal Temperature (°C)	Optimal pH	Thermal Stability	Main Products	Reference
*Vibrio* sp. QY108	CSA/CSB/CSC/CSD	30	7.6	completely inactivated after 40 °C for 1 h	disaccharides	[25]
*Pedobacter heparinus*	DS	30	8.0	ND	disaccharides	[26]
*Flavobacterium heparinum*	CSB	30	8.0	ND	ND	[27]
*Proteus vulgaris*	CSA/CSB	37	7.4	ND	ND	[28]
*Microbulbifer* sp. ALW1	CSB	40	8.0	49.7% residual activity after 60 °C for 2 h	disaccharides	This study

ND: not detected.

## Data Availability

Not applicable.

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
