# Peer review of "Characterization of a Thermostable and Surfactant-Tolerant Chondroitinase B from a Marine Bacterium Microbulbifer sp. ALW1"

_ijms, 2022, doi:10.3390/ijms23095008_

Round 1

Reviewer 1 Report

The MS describes a novel chondroitinase. However, it should be clarified if the closest homolog identity of 87.28% is for a published or putative enzyme (alginate lyase from Microbulbifer sp. HZ11). It should be mentioned how much this enzyme is similar to the published enzymes of this family in terms of protein identity values.

Several issues should be resolved.

1) line 76 "protein with 86.2 kDa was exclusively"

The PAGE indicates that you have an additional band with a MW of 50 kDa. The word "exclusively" is very inaccurate in this context. Is it a degradation product or an impurity? The purification strategy must be mentioned unambiguously, at least in the MM section.

2) A coverage of 35% is not a good result.It must be explained.

3) What does the stability in % indicate? Why is it 80% in the best conditions? A detailed explanation of Fig. 3D must be provided in the text or figure legend.

4) The method describing antioxidant activity should be provided. The authors refer to the previous studies. However, a short explanation should be provided. What kind of chemical reaction mediates antioxidant activity? Were any intermediates/products detected? All this should be clarified, and the reaction scheme should be provided in the MS.

Author Response

Reviewer 1:

Comments and Suggestions for Authors

The MS describes a novel chondroitinase. However, it should be clarified if the closest homolog identity of 87.28% is for a published or putative enzyme (alginate lyase from Microbulbifer sp. HZ11). It should be mentioned how much this enzyme is similar to the published enzymes of this family in terms of protein identity values.

Several issues should be resolved.

Reply: Thank you very much for your comments. We checked the information and found that WP_197023865.1 was a putative alginate lyase from Microbulbifer sp. HZ11. On line 66 in the revised manuscript, “an alginate lyase” is changed to “a putative alginate lyase”.

We added the sequence similarity analysis between ChSase B6 and the published chondroitinase of PL6 family. On lines 68-70 in the revised manuscript, “The sequence similarity between ChSase B6 and the characterized chondroitinase from Pedobacter heparinus of PL6 family was 25.35%.” is added.

1) line 76 "protein with 86.2 kDa was exclusively"

The PAGE indicates that you have an additional band with a MW of 50 kDa. Is it a degradation product or an impurity? The purification strategy must be mentioned unambiguously, at least in the MM section.

Reply: Thank you very much for your comments. The word "exclusively" is indeed inaccurate in this context, and we have deleted it in the revised manuscript (lines 77-78). In order to check the purity of the recombinant protein, we purified the protein with optimized purification strategy. We obtained a single band shown in the revised figure 2A. We have added the purification method on lines 274-284 of the revised manuscript.

2) A coverage of 35% is not a good result. It must be explained.

Reply: Thank you very much for your comments. High-resolution tandem mass spectrometry (MS/MS) has been widely used to identify peptides, and proteins identified by MS/MS can be reconstructed from the peptide data. Although the coverage of 35% is indeed not too high, it is enough to confirm the authenticity of the protein identity. The coverage result is similar to the result (36%) of laminarinase identification by MS/MS (Hu et al., Int. J. Food. Sci. Tech.2021, 56, 4129-4138.).  

3) What does the stability in % indicate? Why is it 80% in the best conditions? A detailed explanation of Fig. 3D must be provided in the text or figure legend.

Reply: Thank you very much for your comments. In our study, the effect of pH on enzyme stability was monitored by scoring the residual activity after incubation the enzyme under varying pH at 37°C for 1 h. The enzyme activity without pH treatment was defined as 100%. The method was described in “3.6 Enzyme property assay”. Therefore, the residual activity was expressed as the relative activity (%). According to the method, the enzyme activity without pH treatment was defined as 100%, the residual activity was determined to be about 80% after treatment under pH 8.0 at 37°C for 1 h. In order to express more clearly, on lines 315 and 319, a detailed explanation has been added in the text of the revised manuscript. On lines 128-131, a detailed explanation has been added in the legend of Fig. 3B and 3D.

4) The method describing antioxidant activity should be provided. The authors refer to the previous studies. However, a short explanation should be provided. What kind of chemical reaction mediates antioxidant activity? Were any intermediates/products detected? All this should be clarified, and the reaction scheme should be provided in the MS.

Reply: Thank you very much for your comments. We have added the hydroxyl radical scavenging assay method on lines 339-346 of the revised manuscript.

Reviewer 2 Report

This manuscript covers characterization of Chondroitinase B from Microbulbifer sp.. The content is interesting and it provides a notable addition to existing body of literature. There are several issues to address:

Major concerns:

Tryptic-digestion and peptide analysis are excellent data to support the predicted molecular mass of ChSase B6. The authors need to elaborate on why this experiment is necessary. Given that gene sequence and predicted amino acid sequences are available, what is the rationale behind the MS analysis? Is it common for post-translational modification of ChSase family enzymes?? The rationale for sequencing is missing.

Authors studied enzyme activity under different ion concentrations. The rationale is missing for the selection of these particular ions. Also, a control experiment using ion-free assay is necessary as a reference. Usually divalent ions such as Mg2+ is preferred for many enzymes, this enzyme seems to operate well with monovalent ion.

Mass peaks for CSB degradation should be labeled (undigested one and digestion products along with their proposed structures/literature ref to support this degradation)

 Minor comments:

Page 1, Line 14:

 a novel chondroitinase B of PL6 family>>> a new member of chondroitinase B of PL6 family (the enzyme is not entirely novel, it is more precise to say a new member of PL6 family)

Page 1, line -18/19:  ChSase B6 demonstrated good thermostability under 60ºC for 2 h with about 18 (remove the word “good”)

Page 3, line 82: These results imply ChSase B6 is a chondroitin sulfate B which specifically degrades CSB and HA (remove this sentence)

Page 4, line 98: The plotting of enzyme activity over the incubation temperature demonstrated that ChSase B6 had an optimum temperature of 40°C>>>>>The plotting of enzyme activity over the incubation temperature demonstrated that ChSase B6 displayed its optimal activity at 40°C –

Page 4, Line 14: Compared to the thermal stability of Chondroitinase ABC from marine bacterium Vibrio sp. QY108 [24], ChSase B6 showed outperformance (Table 2). These…> ChSase B6 exhibited a superior thermostability compared to marine Vibrio sp. QY108-derived Chondroitinase ABC.

Page 4, line 110: These results indicated that 110 ChSase B6 could be active at a relatively wide range of temperature with relatively good 111 thermal stability. (remove this sentence)

Page 5, line 14: Enzyme activity assay for ChSase B6 under different pH conditions showed that  ChSase B6 had optimum pH of 8.0>>> Enzyme activity assay for ChSase B6 under different pH conditions showed that ChSase B6 had optimum activity at pH 8.0

Page 5, line 118: When the pH condition was very alkaline toward pH 11.0, the pH stability of the enzyme decreased abruptly, with 27.8% residual 119 activity (Figure 3D).>>>> When the pH of the reaction solution increased to pH 11.0, the enzyme activity diminished abruptly, with 27.8% residual activity (Figure 3D)

Author Response

Reviewer 2:

Comments and Suggestions for Authors

This manuscript covers characterization of Chondroitinase B from Microbulbifer sp.. The content is interesting and it provides a notable addition to existing body of literature. There are several issues to address:

Major concerns:

Tryptic-digestion and peptide analysis are excellent data to support the predicted molecular mass of ChSase B6. The authors need to elaborate on why this experiment is necessary. Given that gene sequence and predicted amino acid sequences are available, what is the rationale behind the MS analysis? Is it common for post-translational modification of ChSase family enzymes?? The rationale for sequencing is missing.

Reply: Thank you very much for your comments. Tryptic-digestion and peptide analysis can be used to confirm the authenticity of the protein identity. The rationale for protein sequencing by MS has been added in the revised manuscript. On lines 289-292, “Protein identification was performed with Protein Pilot™ v4.5 software (AB SCIEX, USA) against the Malus×domestica database (http://www.rosaceae.org) using the Paragon Algorithm and searching against the genome database of Microbulbifer sp. ALW1 [41].” is added. On lines 77-80, “High-resolution tandem mass spectrometry (MS/MS) has been widely used to identify peptides, and proteins identified by MS/MS can be reconstructed from the peptide data [23].” is added.

Authors studied enzyme activity under different ion concentrations. The rationale is missing for the selection of these particular ions. Also, a control experiment using ion-free assay is necessary as a reference. Usually divalent ions such as Mg2+ is preferred for many enzymes, this enzyme seems to operate well with monovalent ion.

Reply: Thank you very much for your comments. We have no rationale for the selection of the metal ions, and we just selected the commonly assayed ions in the enzyme characterization. We have included the control experiment which is described on lines 325-326 in the revised manuscript. We displayed the results of the control in the revised Figure. 4.

Metal ions might be actively involved in regulating enzyme catalytic action or coordinating enzyme conformational shape during the interaction with substrate, thus affecting enzyme activity. Metal ions, such as Mg2+ and monovalent ion, may have different influences for different enzymes.

Mass peaks for CSB degradation should be labeled (undigested one and digestion products along with their proposed structures/literature ref to support this degradation)

Reply: Thank you very much for your comments. We have labeled the mass peaks for CSB degradation products with the proposed structure and literature ref on lines 169-171 of the revised manuscript. In the revised Figure. 5A, the proposed structure has also been added.

Minor comments:

Page 1, Line 14:

 a novel chondroitinase B of PL6 family>>> a new member of chondroitinase B of PL6 family (the enzyme is not entirely novel, it is more precise to say a new member of PL6 family)

Reply: Thank you very much for the comments. It has been changed accordingly on line 14 in the revised manuscript.

Page 1, line -18/19:  ChSase B6 demonstrated good thermostability under 60ºC for 2 h with about 18 (remove the word “good”)

Reply: Thank you very much for the comments. The word “good” has been deleted in the revised manuscript.

Page 3, line 82: These results imply ChSase B6 is a chondroitin sulfate B which specifically degrades CSB and HA (remove this sentence)

Reply: Thank you very much for the comments. The sentence has been deleted in the revised manuscript.

Page 4, line 98: The plotting of enzyme activity over the incubation temperature demonstrated that ChSase B6 had an optimum temperature of 40°C>>>>>The plotting of enzyme activity over the incubation temperature demonstrated that ChSase B6 displayed its optimal activity at 40°C –

Reply: Thank you very much for the comments. It has been changed accordingly on lines 101-102 in the revised manuscript.

Page 4, Line 14: Compared to the thermal stability of Chondroitinase ABC from marine bacterium Vibrio sp. QY108 [24], ChSase B6 showed outperformance (Table 2). These…> ChSase B6 exhibited a superior thermostability compared to marine Vibrio sp. QY108-derived Chondroitinase ABC.

Reply: Thank you very much for the comments. It has been changed accordingly on lines 112-113 in the revised manuscript.

Page 4, line 110: These results indicated that 110 ChSase B6 could be active at a relatively wide range of temperature with relatively good 111 thermal stability. (remove this sentence)

Reply: Thank you very much for the comments. The sentence has been deleted in the revised manuscript.

Page 5, line 14: Enzyme activity assay for ChSase B6 under different pH conditions showed that  ChSase B6 had optimum pH of 8.0>>> Enzyme activity assay for ChSase B6 under different pH conditions showed that ChSase B6 had optimum activity at pH 8.0

Reply: Thank you very much for the comments. It has been changed accordingly on lines 114-115 in the revised manuscript.

Page 5, line 118: When the pH condition was very alkaline toward pH 11.0, the pH stability of the enzyme decreased abruptly, with 27.8% residual 119 activity (Figure 3D).>>>> When the pH of the reaction solution increased to pH 11.0, the enzyme activity diminished abruptly, with 27.8% residual activity (Figure 3D)

Reply: Thank you very much for the comments. In this study, the effect of pH on enzyme stability was monitored by scoring the residual activity after incubation the enzyme under varying pH at 37°C for 1 h. The enzyme activity without pH treatment was defined as 100%. Fig. 3D displayed the result of the pH stability. Therefore, the expression of “When the pH of the reaction solution increased to pH 11.0” is not so appropriate. The sentence is changed to “When the treatment pH increased to 11.0, the enzyme activity diminished abruptly, with 27.8% residual activity (Figure 3D).” on lines 119-121 in the revised manuscript.